# Effects of structured exercise programmes on physiological and psychological outcomes in adults with inflammatory bowel disease (IBD): A systematic review and meta-analysis

Katherine Jones[1,2]*, Rachel Kimble[3], Katherine Baker[2], Garry A. Tew[1,4]

**1** Warwick Clinical Trials Unit, Warwick Medical School, University of Warwick, Coventry, United Kingdom, **2** Department of Sport, Exercise and Rehabilitation, University of Northumbria at Newcastle, Newcastle Upon Tyne, United Kingdom, **3** Division of Sport and Exercise Science, School of Health and Life Sciences, University of the West of Scotland, Blantyre, United Kingdom, **4** York St John University, Lord Mayor's Walk, York, United Kingdom

* katherine.h.jones@warwick.ac.uk

**Data Availability Statement:** All relevant data are within the manuscript and its Supporting Information files.

## Abstract

### Background

Exercise has been suggested to counteract specific complications of inflammatory bowel disease (IBD). However, its role as a therapeutic option remains poorly understood. Therefore, we conducted a systematic review and meta-analysis on the effects of exercise in IBD.

### Methods

Five databases (MEDLINE, Embase, CINAHL, CENTRAL and SPORTDiscus) and three registers (Clinicaltrials.gov, WHO ICTRP and ISRCTN) were searched from inception to September 2022, for studies assessing the effects of structured exercise of at least 4 weeks duration on physiological and/or psychological outcomes in adults with IBD. Two independent reviewers screened records, assessed risk of bias using the Cochrane Risk of Bias (RoB 2.0) and ROBINS-I tools, and evaluated the certainty of evidence using the GRADE method. Data were meta-analysed using a random-effects model.

### Results

From 4,123 citations, 15 studies (9 RCTs) were included, comprising of 637 participants (36% male). Pooled evidence from six RCTs indicated that exercise improved disease activity (SMD = -0.44; 95% CI [-0.82 to -0.07]; p = 0.02), but not disease-specific quality of life (QOL) (IBDQ total score; MD = 3.52; -2.00 to 9.04; p = 0.21) when compared to controls. Although meta-analysis could not be performed for other outcomes, benefits were identified in fatigue, muscular function, body composition, cardiorespiratory fitness, bone mineral density and psychological well-being. Fourteen exercise-related non-serious adverse events occurred. The overall certainty of evidence was low for disease activity and very low for HRQOL as a result of downgrading for risk of bias and imprecision.

**Funding:** This work was funded through PhD studentship at Northumbria University. The funders had no role in study design, data collection and analysis, decision to publish, or preparation of the manuscript.

**Competing interests:** KJ, GT and KB were investigators on one if the included trials (Jones et al., 2020) and GT on another (Tew et al., 2019), however we do not believe that this has biased our assessment of this or any other study. This does not alter our adherence to PLOS ONE policies on sharing data and materials.

## Conclusions

Structured exercise programmes improve disease activity, but not disease-specific QOL. Defining an optimal exercise prescription and synthesis of evidence in other outcomes, was limited by insufficient well-designed studies to ascertain the true effect of exercise training. This warrants further large-scale randomised trials employing standard exercise prescription to verify this effect to enable the implementation into clinical practice.

## Registration

This systematic review was prospectively registered in an international database of systematic reviews in health-related research (CRD42017077992; https://www.crd.york.ac.uk/prospero/).

## Introduction

Between 25–60% of adults with inflammatory bowel disease (IBD) experience extraintestinal manifestations beyond the gastrointestinal tract, most commonly musculoskeletal manifestations such as osteoporosis, osteopenia, and muscle degeneration [1–3]. The exact mechanisms that may underpin the relationship between osteoporosis and osteopenia and IBD are yet to be fully elucidated. One hypothesised contributing factor is the elevated proinflammatory osteoclast activators (interleukin [IL]-1, IL-6, TNF-α) that interfere with the pathway involved in bone metabolism, formation, and resorption [4, 5]. In addition to malabsorption, vitamin D deficiency, poor calcium intake, reduced physical activity and corticosteroids, which impair osteoblast function, induce osteoblast apoptosis, reduce intestinal calcium absorption, and increase renal excretion of calcium [6–8].

Corticosteroids are also thought to impair muscular function by suppressing protein synthesis, where protein is produced to repair muscle damage, causing a protein imbalance that results in muscle wasting [9–11]. Vitamin D deficiencies also contribute to muscle wasting as it is critical in the mediation of myogenesis, the formation of muscle tissue [12]. Another IBD-specific risk is the elevation of thiobarbituric acid reactive substances and decreased circulating insulin-like growth factor 1 levels, that interfere with the P13K/AKT signalling pathway the is involved in inducing skeletal muscle hypertrophy [13, 14].

Although the presenting symptoms of IBD are mostly physical, the variability between active and quiescent disease states, medication/surgical side effects, symptoms such as fatigue and social and financial repercussions all contribute to an increased risk of depression and anxiety and further profoundly affect a person's quality of life (QOL) [15, 16]. It is understandable that fatigue and QOL worsen during periods of active gut inflammation, however for it to persist in more than 50% of individuals during periods of remission has substantial costs to the individual and on the health care system [17, 18].

An increasing number of studies suggest that structured exercise programmes may be an effective non-pharmacological treatment option to counteract aforementioned manifestations and complications. During exercise an osteogenic stimulus occurs, in which bone is subjected to forces induced by gravitational loading and muscle loading. This osteogenic stimulus initiates an adaptive response involving osteocytes that transduce the energy from the mechanical forces into biological signals that impact bone formation and resorption. This elicits bone deformation, stimulating the stretch-activated ion channel on osteocytes and triggers the

expression of genes that mediate bone growth and increase the threshold of stress tolerance, thus eliciting an architectural modification [19, 20]. Evidence also suggests that increasing skeletal muscle hypertrophy and working capacity during exercise can produce muscle strength and endurance gains through increasing muscle fibre cross-sectional area and oxidisation of substrates to produce adenosine triphosphate [21–23].

Although exercise represents as a line of treatment and has been recognised to counteract secondary complications in other chronic conditions [24, 25], it remains poorly understood. Since the last systematic review in this field of research [26], there have been several exercise interventions published [27–33], four of which were randomised controlled trials [27, 28, 31, 32]. It is important to evaluate non-pharmacological interventions that target these secondary complications, thus avoiding medication side effects and reducing the financial burden on healthcare systems. In doing so, it will allow the integration of the best evidence available to inform evidence-based recommendations to counteract these IBD-specific complications. Therefore, the aim of this review was to evaluate and synthesise the evidence examining the effects of any mode of structured exercise of at least 4 weeks' duration on physiological and psychological outcomes in adults with IBD.

## Methods

The current systematic review has been reported in accordance with the reporting guidance provided in the Preferred Reporting Items for Systematic Reviews and Meta-Analyses (PRISMA) checklist, flow diagram and Explanation and Elaboration document [34]. The Cochrane Handbook was also used to guide the conduct of the review [35]. Deviations from the registered study protocol are detailed in S1 Table.

### Eligibility criteria

Experimental studies such as randomised controlled trials (RCT) and non-RCT's and observational studies assessing the effects of any mode, intensity and frequency of exercise training, of at least 4 weeks' duration were included. Trials comparing one form of exercise versus another, a non-exercising control, usual care group or were uncontrolled were included. Unpublished trials and conference abstracts were only included if the methodological descriptions were provided, either in written form or by direct contact with the authors. Participants had to be aged over 18 years with a clinical diagnosis of IBD (Crohn's disease [CD], ulcerative colitis [UC] or IBD-unclassified) of at least 6 weeks' duration. There were no restrictions on the exercise intervention setting or mode of supervision (e.g., hospital, home-based, unsupervised, supervised). Studies were restricted to English, pertaining to human participants, and must have reported one of the following outcomes for inclusion: bone health, muscular function, QOL, psychological well-being, disease activity, physical activity, body composition, cardiorespiratory fitness, immune function, fatigue, safety, feasibility, and acceptability. A detailed list of outcome measures can be found in S1 File. Where studies reported on multiple measures for one outcome, both measures were included. No outcome measures were prioritised. Where outcomes were measured but not reported, authors were contacted, and data requested.

### Search strategy

Electronic databases from inception to September 2022 were searched: MEDLINE (Ovid), EMBASE (Ovid), CINHAL, Cochrane Central Register of Controlled Trials (CENTRAL) and SPORTDiscus. The search strategy and MESH terms used included: 'inflammatory bowel diseases', 'Crohn disease', 'ulcerative colitis', 'physical activity', 'exercise', 'exercise therapy', 'sports', 'resistance training', 'endurance training', 'aerobic training' and 'physical fitness'. The

full strategy can be found in S2 File. Other relevant completed and ongoing studies were also sought through screening of trial registries (clinicaltrials.gov, WHO International Clinical Trials Registry Platform [ICTRP], International Standard Randomised Controlled Trial Number Registry [ISRCTN]) and forward and backward citations of included studies.

All searches were carried out by the same author (KJ) and search results generated by the electronic databases were exported to EndNote (V8.2), where duplicates were removed. The first 10% of abstracts and titles were examined independently by two review authors (KJ and RK) and due to good agreement (Cohen's $k$ = 0.884) the remaining texts were screened by one reviewer (KJ) [36]. Full text screening was conducted independently by two reviewers (KJ and RK) who recorded reasons for exclusion [35]. Discrepancies were discussed with a third author (GT or KB) and resolved by consensus. Review authors were not blinded to the author, institution, or the publication source of the study.

## Data extraction

The 'Cochrane Data Collection Form for Interventions: RCTs and non-RCT's' was used to extract and record information [35]. Data extracted included: (a) general information such as title, author and publication type; (b) trial characteristics such as study aim, study design, sample size, unit of allocation and duration of participation; (c) participant details such as setting, inclusion/exclusion criteria, method of recruitment, total number randomised, baseline imbalances, withdrawals, age, gender, disease severity and other sociodemographic information; (d) intervention and comparison group(s) characteristics such as number per group, theoretical basis, mode/ frequency/duration/ intensity, progression model, delivery (individual or group based), level of supervision, provider, economic information, resources given, integrity of delivery and compliance; (e) outcome measures such as time points, outcome definition, person measuring, unit of measurement, scales, imputation of missing data, assumed risk estimate and power; and (f) data and analysis such as changes from baseline/time point, reason for missing data, unit of analysis, statistical methods and the possibility of reanalysis.

One review author (KJ) independently extracted data from the included studies with a second review author (RK) independently checking the data extraction forms for accuracy and completeness, with disagreements resolved through consensus of a third review author (GT or KB). Data were entered in Review Manager (RevMan V5.3 Cochrane Collaboration, Oxford, UK) by one reviewer (KJ) and random checks on accuracy were performed by the second reviewer (RK), who kept a record of any discrepancies. Where data were missing or unclear, the corresponding author(s) was contacted via email, at least twice, and relevant information was requested.

## Data synthesis and analysis

For continuous data, the mean and standard deviation change from baseline were imputed to obtain the overall effect size, represented by standardised mean difference (SMD), 95% confidence interval and p value, with <0.05 considered significant. Where change from baseline standard deviations were not reported, but baseline and final standard deviations were, the following equation was used in accordance with the Cochrane Handbook (Section 6.5.2.8): $SD_{change} = \sqrt{SD^2_{baseline} + SD^2_{final} - (2 \times \text{correlation coefficient} \times SD_{baseline} \times SD_{final})}$. Where studies used a different assessment tool to measure the same construct, the standardised mean difference and corresponding 95% confidence interval was calculated. Where studies had multiple assessment time points, data were extracted for the latest time point. To manage multi-arm parallel-groups that compared continuous outcomes of two active arms against a control, these were combined into one active arm for a single pair-wise comparison

using the Review Manager calculator in accordance with Cochrane Training [37]. We planned to include cross-over trials if data were available from the first phase of the study (i.e. before cross-over, as if the trial were running parallel) were available.

Statistical heterogeneity and consistency were assessed by calculating $I^2$, and the following thresholds identified: 0–40% may not be important, 30–60% may represent moderate heterogeneity, 50–90% may represent substantial heterogeneity, 75–100% considerable heterogeneity [35]. If statistical heterogeneity was noted ($I^2$>40%), a random-effects model was used for analysis to account for expected heterogeneity between studies. Sensitivity analysis was performed if considerate heterogeneity ($I^2$>75%) was detected. A sensitivity analysis, assessing the impact of changing the assumptions made, was undertaken where missing data had been imputed. Sensitivity analysis was performed by omitting one study at a time to evaluate the potential bias and robustness of the overall effect estimate.

A narrative summary of adverse events and adherence data is presented. Where data aggregation was not possible due to methodological heterogeneity, these results were summarised narratively.

## Certainty of evidence

Certainty of evidence was analysed independently by two review authors (KJ and RK) using the Grades of Research, Assessment, Development and Evaluation (GRADE) approach, assessing studies against the principal domains: study design, consistency of effect, imprecision, indirectness, inconsistency and publication bias to calculate the overall certainty: high, moderate, low or very low [38, 39].

## Risk of bias

The internal validity and methodological rigor of included studies were assessed independently by two review authors (KJ and RK) using the revised Cochrane 'Risk of Bias' tool (RoB 2.0) for RCTs, the Cochrane Risk of Bias tool for randomised crossover trials and the 'Risk of Bias In Non-randomised studies-of Interventions' (ROBINS-I) for non-randomised studies [40–42]. Discussion between the two review authors was utilised to resolve any discrepancies in judgment of RoB or justifications for judgement, with a third author (GT or KB) referred to in the case of any unresolved discrepancies. The RoB 2.0 addressed five domains including randomisation process, deviations from intended intervention(s), missing outcome data, measurement of outcome and reported results. An additional domain was addressed for the Cochrane ROB tool for randomised crossover trials regarding bias arising from period and carryover effects. Each domain was judged as 'low', 'some concerns', or 'high' risk. The ROBINS-I addressed seven domains including confounding, participant selection, classification of interventions, deviations from intended intervention(s), missing data, measurement of outcomes and reported results. Each domain was judged as 'low', 'moderate', 'no information', 'serious' or 'critical' risk. We derived the overall risk of bias judgement from the highest classified domain i.e. if one domains were classified as high risk this was deemed high risk overall.

## Results

### Search results

The PRISMA flow chart depicting through the review is shown in Fig 1. The search strategy for studies published from inception to September 2022 yielded a total of 4,461 studies, and 3,772 studies after the exclusion of duplicates. Of these, 3,718 were excluded based on the title and abstract, leaving 54 full-text articles to be assessed for eligibility. After full text screening

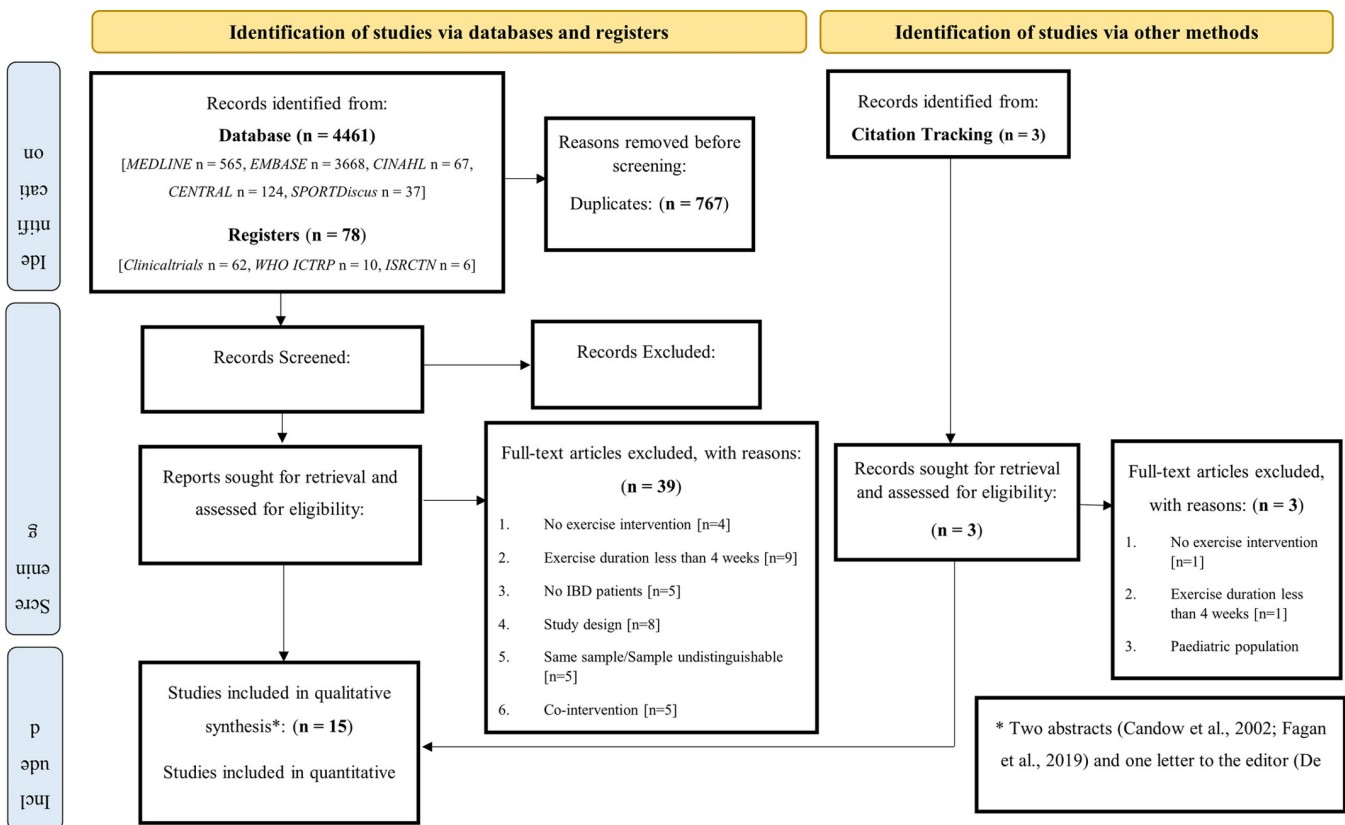

**Fig 1. PRISMA flow diagram of literature search and study selection phases.** n, number; CENTRAL, Cochrane Central Register of Controlled Trials; WHO ICTRP, World Health Organisation International Clinical Trials Registry Platform.

39 studies were excluded, leaving a total of 15 studies included in this review and 6 included in the meta-analysis.

Reasons for exclusion from the meta-analysis were: uncontrolled trials (n = 6) [29, 30, 33, 43–45], outcome measures (n = 2) [46, 47] and a cross-over RCT design (n = 1) [27]. The latter study was excluded from the meta-analysis as guidance for incorporating cross-over trials into a meta-analysis could not be followed, in accordance with the Cochrane Handbook (Section 16.4.5), for the subsequent reasons: (1) it could not be demonstrated that the results approximate those from a paired analysis, as data was reported as median and interquartile ranges and not listed as a method of analysis for cross-over trials (Section 16.4.4), (2) a second approach including only data from the first period could not be performed as this was not reported and (3) to impute standard deviations to attempt to approximate a paired analysis was not achievable as only medians had been reported, which are often use because the data are skewed (Section 7.7.3.5).

Standard deviation data were requested for one study [48], but not available. Therefore, in accordance with the Cochrane Handbook (Section 16.1.3.1) the simplest imputation method was followed, where standard deviation values from two other studies [32, 49] in the same meta-analysis were used, a method shown to yield approximately correct results in two case studies [50]. In addition, synthesis of evidence in other outcomes: bone health, muscular

function, psychological well-being, body composition, cardiorespiratory fitness, immune function and fatigue was unsuitable due to the high levels of heterogeneity and the small number of studies reporting these variables [51].

## Characteristics of included studies

Characteristics of the included studies are summarised in Table 1. Of the fifteen n included studies, published between 1998 to 2022, nine were RCTs [27, 28, 31, 32, 46–49, 52] and five were uncontrolled before and after studies [29, 30, 33, 43–45]. Most studies were conducted in Germany [31, 49, 52] and United Kingdom [28, 32, 46]. Included studies comprised of 637 participants (36% males), with the sample size ranging from 9 to 82 per intervention, 0 to 57 per control group and from 9 to 107 per study. Seven studies [27, 29–31, 33, 45, 47, 52] included both CD and UC participants, six [31, 32, 43, 44, 46, 48] included CD participants only and one [49] included UC participants only. Study participants were described as in remission [27, 30, 47, 49], with an inactive to mildly active disease [28, 29, 31–33, 43, 48, 52], mild to moderately active disease [46] and two studies did not specify disease status [44, 45].

## Exercise interventions

**Exercise programme detail.** The type, intensity, frequency and duration of the structured exercise interventions delivered varied across studies and included: a walking intervention (n = 2) of a low intensity [48] and unquantified intensity [43], a combined aerobic and resistance programme (n = 3) of either a moderate intensity [27, 31] or a moderate to high intensity [30], a resistance training programme (n = 4) of either a low intensity [46], a high intensity [32], or unquantified intensity [44, 45], a low-intensity yoga intervention (n = 3) [33, 47, 49], a moderate-intensity outdoor running programme (n = 1) [52], a moderate-intensity cycling programme (n = 1) [28], and a high-intensity cycling programme (n = 1) [28]. One study did not report the type or intensity of exercise delivered [47]. Nine studies progressed exercise intensity [27, 28, 30–32, 43, 45, 46, 49], determined by the instructor [28, 32], percentage of maximum heart rate [30, 31, 43], based on 1 repetition maximum [30, 45], based on peak power output [28], using the Resistance Intensity Scale for Exercise (RISE) [32] or using the Rating of Perceived Exertion (RPE) Scale [27]. Three studies did not explain how exercise was progressed [33, 46, 49]. Study duration ranged between 8 weeks and 12 months, with a mean duration of 15 weeks and a frequency of sessions ranging from 1 to 5 a week that lasted between 10 and 90 minutes.

**Setting and delivery.** Most studies involved home-based unsupervised sessions [29, 31, 46–48], with two studies [46, 47] providing an initial introductory session to demonstrate correct techniques. Supervision was provided in five studies [27, 30, 43, 45, 52], with one study encouraging home participation only if they could not attend supervised sessions [43]. Four studies delivered a supervised and unsupervised intervention [28, 32, 33, 49], with two encouraging unsupervised exercise participation throughout [32, 49] and one encouraging participants to follow a similar exercise regime following the supervised training period [28]. Programmes were delivered in a home-based setting [29, 31, 46–48], University exercise science facility [28, 32, 43], University gym [27] or not specifically reported [30, 33, 49, 52]. The providers of the intervention included study investigators [28, 32, 52], physiotherapists [30], qualified instructors [27, 33, 45, 47, 49] or not reported [46]. Interventions were delivered as group-based [30, 43, 49], one-to-one [27, 28, 32, 52] or not reported [33, 45]. Delivery method or setting was not reported in one study [44].

**Resource requirements.** Six (40%) of the studies required no equipment to deliver the intervention [29, 31, 47, 49, 52], five used specialised exercise machines [27, 28, 30, 44, 45],

**Table 1. Characteristics of included studies.**

| Author/Year | Study Design | Participants | Interventions | Outcome-measures |
|---|---|---|---|---|
| Robinson et al (1998) [46] | RCT | 107 CD participants with mild to moderate disease activity [IG = 60, CG = 57]. 48 males. Mean age 40.7 | IG: 12-month twice-weekly home-based low-impact progressive resistance training programme<br><br>CG: Usual care | BMD (DEXA) at the femoral neck, greater trochanter and lumbar spine taken at baseline and 12 months |
| Loudon et al (1999) [43] | Uncontrolled pilot study | 12 CD inactive or mildly active participants, no controls. 2 males. Mean age 38.5 | 12-week supervised (indoor track) and unsupervised (outdoors) walking programme 3 sessions (20 min per session, leading to 35 min) a week | Stress (IBBSI), HRQOL (IBDQ), disease activity (HBI), aerobic fitness ($\dot{V}O_2$ max), BMI at 1 and 3 months |
| Candow et al (2002) [44] | Conference abstract; Uncontrolled cohort study | 12 CD participants, no controls. 5 males. Age range 34–51 | Resistance training programme consisting of 3 sets, 8–10 repetitions, 12 exercises working at 60–70% of 1RM 3 times a week over 12 weeks | Disease activity (HBI), muscle strength (1-repetition maximum leg press and chest press) at 1 and 3 months |
| Ng et al (2007) [48] | RCT; Stratified randomisation | Inactive or mildly active 32 CD participants [IG = 16, CG = 16]. 14 males. Mean age 38.8 | IG: 3-month independent low intensity walking programme, working at 40% of aerobic capacity for 30 minutes, 3 times a week<br><br>CG: Usual care and asked to maintain their habitual PA levels | HRQOL (IBDQ), stress (IBDSI), disease activity (HBI) at baseline, 1, 2 and 3 months. PA habits (IPAQ-Long) at baseline and 3 months |
| De Souza-Tajiri et al (2014) [45] | Letter to the editor; Uncontrolled pilot study | 19 women with IBD [CD = 10, UC = 9] and quadriceps weakness. No controls | 8-week progressive supervised resistance training programme twice a week lasting 20 minutes | HRQOL (IBDQ) and quadriceps strength (maximal isometric quadriceps strength and quadriceps 1-RM) at baseline and 8-weeks post exercise |
| Sharma et al (2015) [47] | RCT | 87 IBD [CD = 36, UC = 51] participants in clinical remission [IG = 50, CG = 50]. 54 males. Mean age 34.7 | IG: One supervised yoga session, followed by 7 weeks 1 hour daily home-based sessions<br><br>CG: Usual care | Cardiovascular autonomic functions (heart rate variability through ECG), immune markers (ECP, sIL-2R) and anxiety (STAI) at baseline and 2 months. Clinical symptoms (diary) were recorded |
| Klare et al (2015) [52] | RCT | 30 IBD participants [CD = 19, UC = 11] with an inactive to moderate disease [IG = 15, CG = 15]. 8 males. Mean age 41.1 | IG: Supervised moderate intensity outdoor running, 3 times a week for 10 weeks<br><br>CG: Asked to maintain their current lifestyle behaviours and avoid PA exceeding two hours per week | HRQOL (IBDQ), disease activity (CDAI and RI), BMI, inflammatory markers (CRP and FC) and immune parameters (LC) at baseline and 10-week follow-up |
| Cramer et al (2017) [49] | RCT | 77 UC participants [IG = 39, CG = 38] in clinical remission. 19 males. Mean age 45.8 | IG: 12-week traditional hatha yoga programme, 90 minutes per week<br><br>CG: Received two evidence-based self-care books and were offered the same yoga classes at week 24 | HRQOL (IBDQ) and disease activity (CAI) at week 1, 12 and 24 |
| Cronin et al (2019) [27] | RCT partial cross-over trial | 17 IBD participants [CD = 7, UC = 13] in clinical remission. 15 males. Median age of IG = 33 and CG = 31 | IG: Combined progressive aerobic and resistance, 3 times a week for 8 weeks.<br><br>CG: Usual care, followed by exercise phase after 8 weeks | QOL (SF-36), disease activity (HBI and SCI), psychological well-being (HADS, STAI and BDI-II), body composition (DEXA), pro-inflammatory cytokines (IL6, IL8, IL10 and TNF-a), gut microbiome ($\alpha$ and $b$-diversity) at baseline and 8 weeks |
| Tew et al (2019) [28] | Pilot RCT | Inactive or mildly active 36 CD participants [IG1 = 13, IG2 = 12, CG = 11]. 17 males. Mean age of IG1 = 37, IG2 = 38.5 and CG = 25 | IG1: HIIT of ten 1-minute bouts of cycling at 90% of Wpeak interspersed with 1-minute bouts of 15% of Wpeak 3 times a week for 12 weeks<br><br>IG2: MICT of 30 minutes of cycling at 35% Wpeak. 3 times a week for 12 weeks<br><br>CG: Usual care group, offered an exercise consultation following study completion | Feasibility, acceptability, safety, HRQOL (IBDQ), QOL (EQ-5D), Fatigue (IBD-F), psychological well-being (HADS) and physical activity habits (IPAQ) at week 1, 13 and 26. Body mass, waist circumference, blood pressure, resting heart rate, cardiorespiratory fitness (Ventilatory threshold and peak oxygen uptake), disease status (CDAI) and inflammatory status (FC) at week 1 and week 13. |
| Fagan et al (2019) [29] | Conference abstract; Uncontrolled prospective cohort study | 82 IBD participants with an inactive or mildly/moderately active disease. No controls | Home-based personalised exercise programme 5 times a week lasting 10 minutes per session for a total of 4 months | Fatigue (MFI), anxiety, depression and HRQOL (IBDQ), disease activity (HBI, SCI), physical activity (IPAQ) at baseline and 4 months. Blood and stool samples were also analysed. |

*(Continued)*

**Table 1.** (Continued)

| Author/Year | Study Design | Participants | Interventions | Outcome-measures |
|---|---|---|---|---|
| van Erp et al (2021) [30] | Uncontrolled pilot study | 25 inactive IBD participants with severe fatigue ($\geq$3 months) [CD = 21; UC = 3; IBDU = 1]. 15 males. Mean age 45 | Personalised aerobic and progressive-resistance training programme thrice a week for 12 weeks with each session lasting for 1 hour (30 minutes aerobic and 30 minutes resistance). The programme consisted of using an indoor bicycle/cross-trainer/treadmill at 65–80% of maximum HR and 8 resistance training machines performing 15–20 repetitions at a weight of 40–60% of 1RM. | Fatigue (CIS), HRQOL (IBDQ), cardiorespiratory fitness (CPET) with maximum power and maximum oxygen uptake ($\dot{V}O_2$ max), body composition (BMI, BF% [skinfolds], BP and resting heart rate [ECG]) at baseline and 12 weeks |
| Seeger et al (2020) [31] | Pilot RCT | Inactive or mildly active 45 IBD participants [IG 1 = 17, IG 2 = 15, CG = 13]. 17 males. Mean age IG1 = 39.6, IG2 = 42 and CG = 43.7 | IG1: Moderate endurance home-based exercise programme performed for 30 minutes, thrice weekly for 12 weeks | Disease activity (CDAI, PRO2), lower extremity strength (isometric measurement of the quadriceps), handgrip strength (dynamometer), laboratory parameters (FC, TBC) and anthropometric data (BP, pulse, BMI and respiratory rate) at baseline and 3 months. HRQOL (sIBDQ), physical activity levels (sIPAQ), and disease activity (PRO2) at baseline, 3 and 6 months. |
| | | | IG2: Muscular training programme consisting of 2–3 sets, 12 exercises performed three times a week and lasting 30–40 minutes each for 12 weeks | |
| | | | CG: Usual care | |
| Jones et al (2020) [32] | RCT | Inactive or mildly active 47 CD participants [IG = 23; CG = 24]. 15 males. Mean age IG = 46.1, CG = 52.3 | IG: 6-month progressive combined impact and resistance training programme consisting of skipping, 2–3 sets of 10–15 repetitions for 5 multidirectional jumps followed by 2–3 sets of 10–15 repetitions for 8–10 TheraBand® elastic band exercises, three times a week. Programme was primarily homebased with 12 supervised sessions gradually tapered over 6 months. | BMD (DEXA) at the femoral neck, greater trochanter and lumbar spine at baseline and 6 months. Upper body strength (handgrip dynamometer, isokinetic strength of elbow flexors), lower body strength (isokinetic strength of knee extensors). Lower-body muscular endurance (30 seconds chair stand test), upper-body muscular endurance (30 seconds bicep curl test). HRQOL (IBDQ), QOL (EQ-5D-5L), fatigue (IBD-F), physical activity levels (SPAQ), disease activity (CDAI, FC) and anthropometric data (BP and resting heart rate) at baseline, 3 and 6 months |
| | | | CG: Usual care comprised of evidence-based medical treatment and an exercise consultation offered to control participants following completion of the study | |
| Kaur et al (2022) [33] | Uncontrolled Pilot Study | 9 IBD participants with an active (n = 5) or inactive (n = 4) disease [CD = 6; UC = 3]. 1 male. Mean age 52.1 | 8-week combined supervised and home-based yoga intervention. Supervised sessions were delivered once a week lasting 30 minutes, the first session was delivered over two hours. Home-based sessions were completed daily. | Adherence, adverse events, salient beliefs, acceptability, and safety. Depression (PHQ-9), anxiety (GAD-7), perceived stress (PSS-10), HRQOL (SF-12), sleep quality (PSQI) and disease activity (HBI and Partial Mayo Score) at baseline and 8 weeks |

IG, Intervention Group; CG, Control Group; CD, Crohn's Disease; UC, Ulcerative Colitis; HRQOL, Health-Related Quality of Life; BMI, Body Mass Index; IBD, Inflammatory Bowel Disease; IBDQ, Inflammatory Bowel Disease Questionnaire; HBI, Harvey Bradshaw Index; RCT, Randomised Controlled Trial; FC, Faecal Calprotectin; CRP, C-reactive Protein; RI, Rachmilewitz Index; DEXA, Dual-Energy X-Ray Absorptiometry; PA, Physical Activity; IPAQ, International Physical Activity Questionnaire; ECG, Electrocardiogram; CAI, Clinical Activity Index; SCI, Simple Colitis Index; STAI, State and Trait Anxiety Index; BDI-II, Beck Depression Inventory-II; SF-36, Short-Form 36; LC, Leucocytes; sIL-2R, Soluble Interleukin-2 Receptor; ECP, Eosinophilic Cationic Protein; HIIT, High-Intensity Interval Training; MICT, Moderate-Intensity Continuous Training; WPeak, Peak Power Output; MFI, Multidimensional Fatigue Inventory; sIBDQ, Short Inflammatory Bowel Disease Questionnaire; PRO2, Patient Reported Outcome Score 2; sIPAQ, Short International Physical Activity Questionnaire; BP, Blood Pressure; IBDU, Inflammatory Bowel Disease Unclassified; BF%, Body Fat Percentage; ECG, Electrocardiogram; PHQ, Patient Health Questionnaire; GAD, Generalised Anxiety Disorder Assessment; PSS, Perceived Stress Scale; SF-12, Short-Form 12; PSQI, Pittsburgh Sleep Quality Index

heart rate monitors were used in two studies [43, 48] and resistance bands or tubing were used in two studies [32, 46].

**Motivational techniques.** Motivational strategies within the exercise interventions included: written instructional exercise booklets [29, 31, 32, 46, 47, 49] or as a smartphone application [31, 33], support meetings [46], online videos [33], telephone support and motivation calls [32, 47] and goal setting [32].

**Supervision/ adherence.** Thirteen (87%) studies [27–33, 43, 46–49, 52] provided data on adherence which ranged from 52% to 89%. To monitor adherence, six studies employed self-reported logs or diaries [32, 33, 43, 46, 48, 49], three monitored activity through monitoring system [27, 43, 48] and one used a goal chart [29].

Withdrawals were reported in ten studies [27, 28, 31–33, 43, 46, 47, 49, 52]. Out of 343 exercising participants, 51 (14.9%) withdrew due to reasons such as lack of motivation/ loss of interest (n = 13), scheduling issues (n = 12), active disease (n = 5), pregnancy (n = 4), moved away (n = 2), loss of follow up (n = 3) or other (n = 12). Out of 215 control participants, 20 (9.3%) withdrawals were reported for reasons such as loss to follow up (n = 3), active disease (n = 3), uncontactable (n = 2), pregnancy (n = 2), death (n = 2), adverse events (n = 2), no longer interested (n = 2), or other (n = 4). Including only controlled studies that reported withdrawals [27, 28, 31–33, 46, 47, 49, 52], the dropout rate for exercising participants was 17.4% (n = 42/241) compared to 9.3% (n = 20/215) for control participants.

**Control group.** Structured exercise interventions were compared to a control group in nine studies [27, 28, 31, 32, 46–49, 52]. These included standard care only [47] in conjunction with asking participants to maintain their habitual physical activity level [27, 48, 52], offering a telephone-based consultation following final study assessments to discuss individual facilitators/barriers to exercise and how to incorporate physical activity into their lifestyle [28, 32] and providing two self-care books [49]. These books provided information on the disease pathology and pathophysiology, possible diagnostic procedures and treatments with emphasis on self-care strategies such as appropriate lifestyle modification, over-the-counter medication, naturopathic and integrative medicine treatment approaches and physical treatments. Two studies did not specify what control participants received [31, 46].

## Findings by outcome

**Disease activity.** Eleven studies [27–29, 31–33, 43, 44, 48, 49, 52] measured disease activity using comparable indices including the: HBI [27, 29, 31, 43, 44, 48], CDAI [28, 32, 52], SCI [27, 29], CAI [49], PRO-2 [31] and RI [52]. However, only six of these were RCT's [28, 31, 32, 48, 49, 52] with data suitable for inclusion within a meta-analysis (n = 258). The meta-analysis of change in disease activity demonstrated a statistically significant reduction in exercise versus control (SMD -0.46; 95% CI -0.83 to -0.09; p = 0.01), with a moderate heterogeneity ($I^2$ 50%) (Fig 2). Among the studies, the largest effect on disease activity was observed following a low intensity walking programme [48] (SMD -1.58; -2.38 to -0.77). Sensitivity analysis was performed as a result of imputing missing variability data. Although the removal of a single study [48] from the pooled analysis of disease activity data reduced heterogeneity ($I^2$ 0%), the group difference remained statistically significant (SMD -0.29; -0.56 to -0.03; p = 0.03).

Of the studies that could not be included in this meta-analysis, one reported a significant improvement (HBI 5.9 ± 5.0 to 3.6 ± 3.1; p = 0.02) [43], one reported evidence of improvement nevertheless did not achieve statistical significance in CD participants (p = 0.084) but did in UC participants (p = 0.021) [29]. For the remaining three studies, two reported no change [27, 44] and one reported higher disease activity scores using the HBI (5.6 ± 1.5 to 6.2 ± 1.9) and partial mayo scores (0.7 ± 1.2 to 1.3 ± 2.3) [33]. However, as the aim of this study was not to determine intervention effectiveness, no statistical analysis was undertaken.

Inflammatory markers to determine disease activity were also determined in five studies [28, 29, 31, 32, 52]. One assessed C-reactive protein [52], one assessed total blood count [31] and one collected blood and stool samples, but did not report the method of analysis undertaken [29]. Four studies assessed faecal calprotectin [28, 31, 32, 52], however due to the variation in reporting were not included within a meta-analysis. No significant changes in

| Study or Subgroup | Exercise Mean | Exercise SD | Exercise Total | Control Mean | Control SD | Control Total | Weight | Std. mean difference IV, Random, 95% CI |
|---|---|---|---|---|---|---|---|---|
| Cramer et al 2017 | 1.7 | 2.1 | 27 | 2.8 | 2.7 | 34 | 20.5% | -0.44 [-0.95 , 0.07] |
| Jones et al 2020 | -24.8 | 46.5 | 20 | 5.1 | 77.3 | 15 | 15.7% | -0.48 [-1.15 , 0.20] |
| Klare et al 2015 | -11.3 | 38 | 30 | -7.6 | 30 | 30 | 20.7% | -0.11 [-0.61 , 0.40] |
| Ng et al 2007 | 3.6 | 2.1 | 16 | 7 | 2.1 | 16 | 12.9% | -1.58 [-2.38 , -0.77] |
| Seeger et al 2020 | -3.7 | 48.8 | 22 | 9.4 | 60.9 | 13 | 15.5% | -0.24 [-0.93 , 0.45] |
| Tew et al 2019 | 4 | 50.8 | 24 | 15 | 36.7 | 11 | 14.8% | -0.23 [-0.94 , 0.49] |
| **Total (95% CI)** | | | **139** | | | **119** | **100.0%** | **-0.46 [-0.83 , -0.09]** |

Heterogeneity: Tau² = 0.10; Chi² = 9.96, df = 5 (P = 0.08); I² = 50%
Test for overall effect: Z = 2.46 (P = 0.01)
Test for subgroup differences: Not applicable

**Fig 2. Meta-analysis of change in disease activity scores following an exercise intervention in adults with IBD.** SD, Standard Deviation; Std, Standard; IV, Weight Mean Difference; CI, Confidence Interval.

inflammatory markers were observed in any study. In addition, no exercising participant experienced a relapse or significant worsening of inflammatory markers.

**Quality of life.** The effects of an exercise intervention on QOL have been explored in eleven studies which reported data using the disease-specific IBDQ [28–33, 43, 45, 48, 49, 52]. Six studies [28, 31, 32, 48, 49, 52] (n = 271) were included in a meta-analysis that showed no clear difference in change between exercise and control (Total IBDQ score MD 3.52; -2.00 to 9.04; p = 0.21), with a substantial heterogeneity (I² 73%) (Fig 3). Among the studies, the largest effect on QOL were observed following a yoga intervention [49] (MD = 17.50; 5.42 to 29.58). Sensitivity analysis was performed as a result of imputing missing variability data. With the removal of a single study [28], the meta-analysis became statistically significant (MD -0.28; -0.55 to -0.01; p = 0.04) and heterogeneity reduced (I² 66%). Of all the studies that reported data using the IBDQ that could not be included, one reported that 31% of participants achieved clinically meaningful improvements [29] and three reported significant improvements in total IBDQ [30, 43, 45]. Klare and colleagues [52] reported significant improvements in all four IBDQ subdomains and all but bowel function (p = 0.208) in the intervention delivered by van Erp et al. [30].

Of the studies that reported QOL data using the SF-36 [27], SF-12 [33] and the EQ-5D-5L [28, 32]. higher physical and mental component scores (SF-12) and superior EQ-5D-5L

| Study or Subgroup | Exercise Mean | Exercise SD | Exercise Total | Control Mean | Control SD | Control Total | Weight | Mean difference IV, Random, 95% CI |
|---|---|---|---|---|---|---|---|---|
| Cramer et al 2017 | 24.3 | 34.2 | 39 | 6.8 | 17.4 | 38 | 11.4% | 17.50 [5.42 , 29.58] |
| Jones et al 2020 | 4 | 10.3 | 22 | 0 | 12.7 | 21 | 17.8% | 4.00 [-2.93 , 10.93] |
| Klare et al 2015 | 28.3 | 24.5 | 30 | 14.5 | 16.1 | 30 | 13.1% | 13.80 [3.31 , 24.29] |
| Ng et al 2007 | 5.98 | 10.3 | 16 | 5.24 | 10.3 | 16 | 17.5% | 0.74 [-6.40 , 7.88] |
| Seeger et al 2020 | 2.6 | 5.1 | 22 | 1.6 | 3.6 | 13 | 22.9% | 1.00 [-1.89 , 3.89] |
| Tew et al 2019 | 2.7 | 8.7 | 23 | 11 | 10.7 | 11 | 17.3% | -8.30 [-15.55 , -1.05] |
| **Total (95% CI)** | | | **152** | | | **129** | **100.0%** | **3.44 [-2.15 , 9.03]** |

Heterogeneity: Tau² = 33.28; Chi² = 19.69, df = 5 (P = 0.001); I² = 75%
Test for overall effect: Z = 1.21 (P = 0.23)
Test for subgroup differences: Not applicable

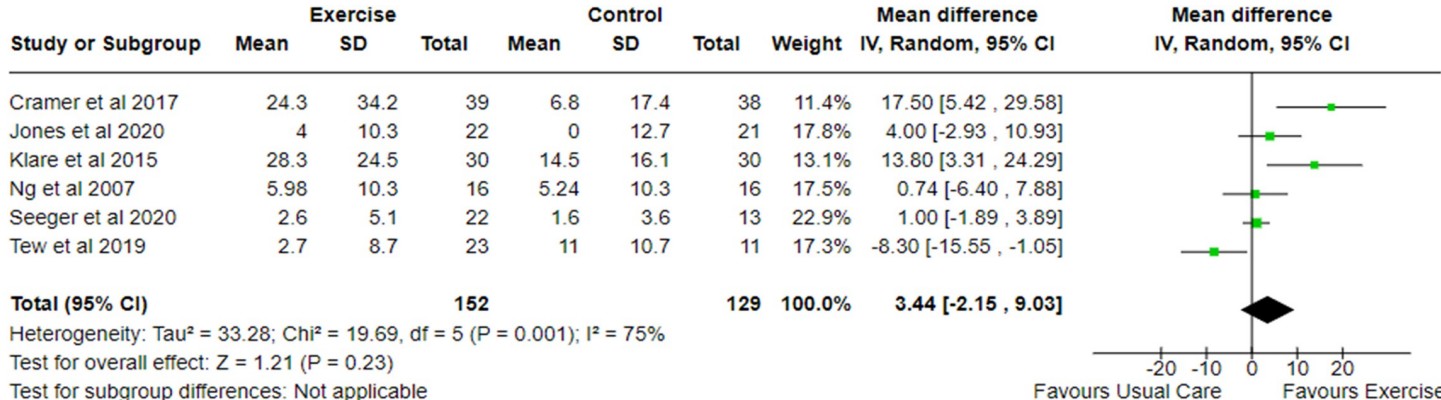

**Fig 3. Meta-analysis of change in QOL following an exercise intervention in adults with IBD using the IBDQ.** SD, Standard Deviation; Std, Standard; IV, Weight Mean Difference; CI, Confidence Interval; IBDQ, Inflammatory Bowel Disease Questionnaire.

outcome data were identified following an uncontrolled yoga intervention and combined impact and resistance training programme, respectively [32, 33]. The remaining two studies reported no change [27, 28].

**Bone health.** Bone mineral density was assessed at the femoral neck, greater trochanter and lumbar spine using DXA in two studies, the first employing a 12-month low impact resistance training programme [46] and the second a 6-month combined impact and resistance training programme [32]. The first study [46] identified those who were fully compliant with the programme (n = 14/53) and demonstrated statistically significant group differences at the greater trochanter ($\Delta$ = 4.67%; 95% CI 0.86 to 8.48; p = 0.02) and greater mean difference improvements at the lumbar spine and femoral neck compared to non-compliant participants. In contrast, Jones et al. [32] identified no significant group differences at the greater trochanter or femoral neck but did identify significant improvements at the lumbar spine (adjusted mean difference 0.036g/cm$^2$; 95% CI 0.024–0.048; p<0.001) following a 6-month intervention.

**Muscular function.** Four studies [31, 32, 44, 45], two of which were uncontrolled, examined the effects of an exercise intervention on muscular function: muscular strength (n = 4) and muscular endurance (n = 1). All studies, which employed an element of resistance training, saw a significant improvement (all p<0.05) in muscular strength in the upper [31, 32, 44] and lower [31, 32, 44, 45] extremities. Only one study explored the effects of exercise on muscular endurance. This 6-month combined impact and resistance training programme in inactive to mildly active CD saw significant improvements in upper and lower limb muscular endurance (both p<0.001), measured by the 30-s bicep curl test and 30-s chair stand test [32].

**Psychological well-being.** The effects of an exercise intervention on psychological well-being has been explored in seven studies [27–29, 33, 43, 47, 48], measures which included: stress (IBD Stress Index [n = 2]; Perceived Stress Scale-10 [n = 1]), anxiety (State and Trait Anxiety Scale [n = 2]; Generalised Anxiety Disorder Assessment-7 [n = 1]) and depression (Hospital Anxiety and Depression Scale [n = 2];Beck Depression Inventory II [n = 1]; Patient Health Questionnaire-9 [n = 1]). One study did not specify the anxiety and depression instrument used [29].

Two interventions aiming at reducing stress saw significant within group improvements following a 12-week walking programme (both p<0.005) [43, 48], however one saw no change in stress following a 8-week yoga programme [33]. Interventions assessing anxiety saw significant reductions following an 8-week home-based yoga programme (p = 0.01) [47] and unsupervised 4-month personalised exercise programme (p<0.05) [29]. Anxiety and depression mean scores also improved in both the HIIT and MICT (anxiety = HIIT 5.5 vs 5.2; MICT 6.8 vs 5.5; depression HIIT = 3.6 to 2.7; MICT = 3.8 to 2.7), however statistical significance was not assessed due to the study focusing on feasibility [28]. Kaur et al. [33] also undertook no statistical analysis after an uncontrolled pilot study, however reported improvements in both mean anxiety (5.4 ± 4.1 vs 4.2 ± 4.0) and depression scores (9.8 ± 6.3 vs 5.4 ± 4.1) after 8 weeks. In contrast to these findings, a randomised controlled cross-over trial exploring the influence of an 8-week combined moderate aerobic and resistance training programme in 17 IBD participants in clinical remission reported no changes in depression or anxiety measured using the HADS, STAI and BDI-II [27].

**Fatigue.** Four studies [28–30, 32], two of which were uncontrolled, have assessed fatigue following an exercise intervention. The study of Tew et al. [28] explored the effects of 12 weeks of HIIT or MICT in adults with inactive to mildly active CD. Following the intervention, the mean change in total fatigue, assessed using the IBD-F from baseline to 3 months and 6 months was 0.1 and -0.7 respectively in the HIIT group and 0.5 and -0.5 respectively in the MICT group, with lower scores representing improving fatigue. However, given the intentionally small sample size the study was underpowered to detect effect and therefore results

should be interpreted with caution. Nevertheless, post exit interviews confirmed 8 participants reported feeling more energised.

Significant improvements were also identified following a 4-month home-based personalised booklet intervention (p<0.05) [29] and 12-week aerobic and progressive resistance training which saw significant improvements in the total CIS (p<0.001) and subdomains: severity (p<0.001), concentration (p = 0.001), motivation (p<0.001) and activation (p = 0.001) [30]. Interestingly, Jones and colleagues [32] 6-month combined impact and resistance training programme saw no significant changes in the IBD-F at 3 months, but did at 6 months (adjusted mean difference −2, 95% CI −4 to −1; p = 0.005) [32].

**Cardiorespiratory fitness outcomes.** Four studies [27, 28, 30, 43], two of which were uncontrolled, have examined the effect of exercise programmes on cardiorespiratory fitness. Using the Canadian Aerobic Fitness Step Test, Loudon and colleagues [43] 12-week walking programme demonstrated significant improvements in maximal aerobic capacity ($\dot{V}O_2$ max) between pre and post measures (30.6 ± 4.7 vs 32.4 ± 4.8; p = 0.0013). A combined progressive resistance and aerobic programme cross-over RCT also identified significant improvements in $\dot{V}O_2$ max estimated from the Rockport one-mile walk test (43.4 mL/kg/min vs 46.0 mL/kg/min; p = 0.03). Similarly, Tew and colleagues [28] feasibility RCT, saw a positive mean change in peak oxygen uptake and ventilatory threshold from baseline to 3 months in the HIIT (27.3 vs 29.7 mL/kg/min; 16.5 vs 16.8 mL/kg/min, respectively) and MICT (28.7 vs 29.3 mL/kg/min; 16.0 vs 18.2 mL/kg/min, respectively) groups. Lastly, although significant within-group improvements were identified in maximum power (p = 0.002) following a 12-week personalised aerobic and progressive-resistance training programme, no significant differences were identified in $\dot{V}O_2$ max (p = 0.077) [30].

**Body composition.** Body composition changes were assessed in five [27, 30, 31, 43, 52] out of fourteen included studies. Firstly, Cronin et al. [27] assessed body fat and lean tissue mass, after a combined moderate aerobic and resistance programme. Following the conclusion of this 8-week crossover RCT, significant improvements, assessed by DXA, were achieved by the exercise group demonstrating a median decrease of 2.1% (-2.15, -0.45) (p = 0.022) body fat and a median increase of 1.59kg (0.68, 2.69) (p = 0.0003) lean tissue mass when compared to the non-exercising cohort.

Body mass index (BMI) was also assessed in four studies [30, 31, 43, 52], two of which were uncontrolled. Although an increase in BMI was identified by Klare et al. [52] and a reduction by Loudon et al. [43] neither of these were statistically significant (both p = 0.07).

**Immune parameters.** Three studies [27, 47, 52] included in this review determined the effects of exercise on immune parameters. However, heterogeneity in the markers obtained precluded a meta-analysis. Across the three studies, no significant changes were identified in proinflammatory cytokines (IL-8, IL-10, IL-6 and TNF-a), a-diversity, taxonomic b-diversity, serum eosinophilic cationic protein, soluble interleukin-2 receptor or leucocyte count. However, although not statistically significant, there was a modest increase in gut microbiota a-diversity following a 8-week combined aerobic and resistance programme [27].

**Physical activity.** Five studies [28, 29, 31, 32, 48] used questionnaires (International Physical Activity Questionnaire [IPAQ] n = 3, short IPAQ n = 1 and the Scottish Physical Activity Questionnaire [SPAQ] n = 1) to evaluate physical activity. Four studies [28, 29, 31, 32] used the measurement as an outcome and the remaining study [48] used the measurement to assess whether habitual activity levels changed in the control group and assess other physical activity habits in the participants' lifestyle aside from the exercise intervention.

Outcome measures of physical activity in two studies demonstrated a significant increase in total number of metabolic equivalents (MET)-min/week [29, 31]. An increase in physical activity levels were identified in the HIIT group at 3 months, but not at 6 months and not at 3

or 6 months in the MICT group in Tew and colleagues [28] feasibility study. No significant between group differences were demonstrated at 3 or 6 months following a 6-month combined impact and resistance training programme using the SPAQ [32].

**Adverse events.**  In total, seven studies [27, 28, 32, 33, 46, 49, 52] (46.7%) reported on adverse events (AE). Two studies reported no AEs occurring [27, 46]. Fifteen exercise-related non-serious adverse events were reported across the other five studies [28, 32, 49, 52] which were: musculoskeletal pain (n = 5) [49], acute flares (n = 2) [52], abdominal pain during and after training (n = 1) [52], light-headiness (n = 2) [32], mild headaches and dizziness (n = 2) [28], vomiting following exercise (n = 1) [28],nausea (n = 1) [32] and skin irritation (n = 1) [33]. The study by Cramer et al. [49] reported three serious adverse events including: two hospital stays due to acute flares and one diagnosis of colorectal cancer. The relationship to the intervention and expectedness was not reported. All other studies reported no serious adverse events.

**Ongoing trials.**  Ongoing clinical trials and unpublished studies were searched on ClinicalTrials.gov, WHO ICTRP and ISRCTN registry. Six studies of interest were identified on ClinicalTrials.gov (NCT04589338; NCT04303260; NCT04143490; NCT02849717; NCT03177044; NCT04273399) and one on the ISRCTN registry (ISRCTN10756924). All, detailed in S2 Table, are exploring the effect of different intervention strategies including a form of physical activity in IBD with a current status of recruiting, not yet recruiting or have had recruitment suspended due to COVID-19. One study of interest (NCT02861053) remains classified as ongoing, with attempts made to obtain an update on the progress, with no success.

**Risk of bias.**  We assessed the risk of bias of every outcome for nine studies using the algorithm guidance from the Cochrane RoB 2.0 and randomised crossover trial tool, and six studies using the ROBINS-I tool (S3 Table). No study outcome was rated as low risk of bias. All outcomes were rated with some concerns or with a high risk of bias, for blinding of participants or personnel to the intervention or outcomes, in which particularly participant-reported outcomes could have been influenced by knowledge of the intervention received. Risk of bias graphs for meta-analytic outcomes are presented in Fig 4 and risk of bias summary plots in S1 Fig.

**Certainty of evidence.**  The certainty of evidence was low for meta-analytic outcome disease activity (6 RCTs, n = 282) and very low for HRQOL (6 RCTs, n = 307) (S4 Table). The main reasons for downgrading the evidence were risk of bias and imprecision, including the 95% confidence interval crossing the line of no effect and the optimal information size not reached for either outcome.

## Discussion

The present study aimed to evaluate and synthesise the evidence concerning any mode of structured exercise of at least 4 weeks' duration on physiological and psychological outcomes in adults with IBD. This comprehensive systematic review includes six new exercise studies [27–32], four of which were RCTs [27, 28, 31, 32] and two novel meta-analyses [28, 31, 32, 48, 49, 52].

In this systematic review and meta-analysis, we report low-certainty evidence of a small-to-moderate improvement in disease activity in people who participated in exercise training compared to those who did not (6 RCTs, n = 282). Exercise did not significantly improve health-related quality of life scores, with no differentiation between groups and very low certainty (6 RCTs, n = 307). Although a meta-analytic summary was not possible for some outcomes there is preliminary evidence of a beneficial effect of exercise on: bone health, muscular function, psychological well-being, fatigue, cardiorespiratory fitness, body composition, immune parameters and inflammatory biomarkers. However, findings should be interpreted with caution due to the substantial heterogeneity within and across studies and inclusion of small pilot or feasibility studies.

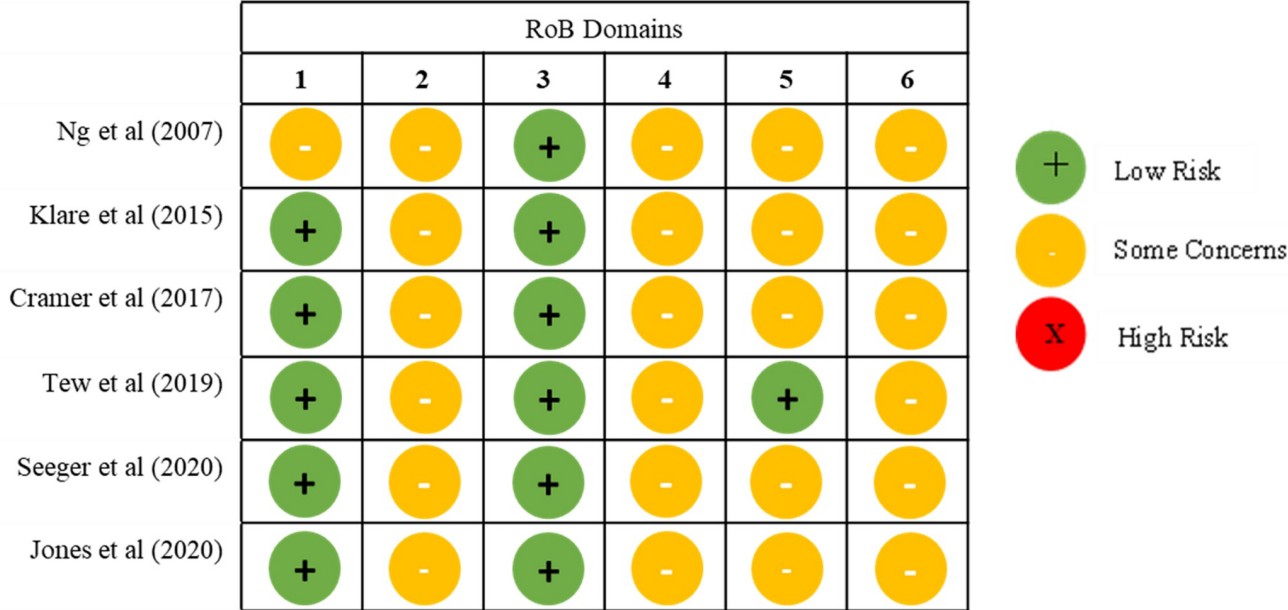

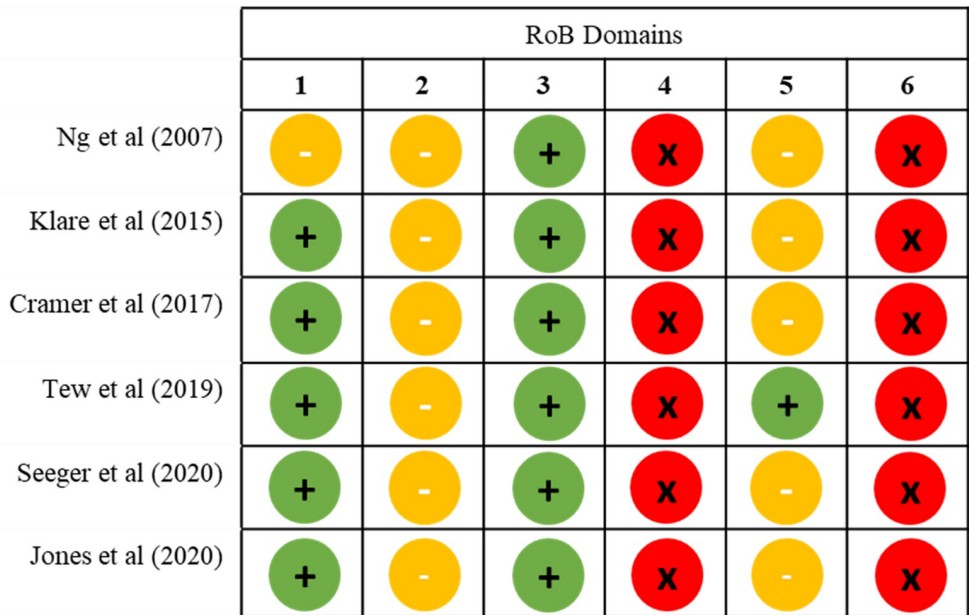

**Fig 4. Risk of bias graph (A) disease activity, (B) health-related quality of life for meta-analysis.** RoB 2.0 Tool Domains: 1) Bias arising from the randomisation process; 2) Bias due to deviations from intended interventions; 3) Bias due to missing outcome data; 4) Bias in measurement of the outcome; 5) Bias in selection of the reported result; 6) Overall bias.

This review differs from the findings of previous reviews who report neither positive nor negative changes in disease activity [1, 26, 53, 54]. A variation that may be explained by the exclusion of studies with less than a 4-week exercising period. Although single bouts, or shorter durations of exercise in IBD have demonstrated increases in albumin, haemoglobin,

erythrocytes, haematocrit and leukocytes [55–57] longer durations have demonstrated to decrease erythrocyte sedimentation rate, C-reactive Protein and thrombocytes, which are considered indicators for a diminished inflammation [1, 55, 58, 59]. In addition, prolonged beneficial effects on the gastrointestinal tract may be as a result of an increased gastrointestinal mobility, thereby reducing the contact time of pathogens within the gastrointestinal mucus layer, which has been identified following longer durations of exercise [1, 53]. In addition, our narrative synthesis of objective markers of inflammation (FC and CRP) indicated no change following an exercise intervention, a finding similar to previous review [26]. This suggests that people with IBD self-report a benefit to their symptoms and are capable of exercising symptom free without experiencing exacerbation of symptoms. These studies are limited by their inclusion of participants with an inactive or mildly active disease only, which left not much potential for improvement. Further trials including participants with a moderate or severely active disease are warranted.

In contrast to previous reviews [1, 26, 53, 54] no improvement was identified in health-related quality of life (HRQOL). This may be a reflection of the limited number of studies included in the meta-analysis, very low certainty of evidence and the heterogeneity of exercise interventions included. After conducting a sensitivity analysis by removing one study, the pooled result revealed a significant improvement in HRQOL and reduced heterogeneity. Of the studies not included within the meta-analysis that determined QOL all but one [27] reported achieving either clinically meaningful improvements or statistical significance following an exercise intervention. On this basis, it seems that the effect of exercise on QOL may be potentially beneficial to adults with an inactive to mildly active disease and therefore consistent with previous findings.

Despite consensus from the European Crohn's and Colitis Organisation [60] addressing that weight-bearing or resistive exercises should be implemented to prevent bone loss, there are no specific clinical guidelines regarding the exercise type, intensity, duration or frequency to elicit favourable changes. Previous reviews have highlighted improvements, although not significant, in bone mineral density at the greater trochanter, femoral neck and lumbar spine following a 12-month low-impact resistance programme [46]. Since the last review in this research field [26] new evidence has been identified which partially contradict the results of previous reviews. A more recent high-impact and resistance training programme found no significant group differences at the greater trochanter or femoral neck at 6 months but did identify significance at the lumbar spine [32]. A possible explanation for this variation between these two RCTs could be explained by the intensity of the interventions offered, with Robinson et al. [46] delivering a low impact programme and Jones et al. [32] delivering a high impact programme. The size of an osteogenic effect, an adaptive response that converts energy from mechanical forces into biological signals that impact bone formation and resorption, relies greatly on the loading frequency and intensity with greater osteogenic effects occurring with higher strains of mechanical loading, induced by gravitational and muscle loading [4, 5, 19]. Given the high risk of developing osteopenia or osteoporosis compared with the general population [7] the potential implications and importance of encouraging adults with IBD to remain active and incorporate impact and weight-bearing exercises is essential to preserve BMD and to support the effective use of NHS resources.

Improvements in muscular strength following an exercise intervention highlighted in this review are in line with previous findings [1, 26, 53, 54]. However, to our knowledge this is the first review to highlight the potential of exercise to significantly improve upper and lower limb muscular endurance. Despite significantly impaired muscular function identified in IBD [61–64], and its association with reduced QOL, morbidity, mortality and further disability in other chronic conditions it remains one of the least researched extra-intestinal manifestations with

IBD [65, 66]. These findings highlight the importance of detecting muscle weakness in people with IBD as part of routine clinical care and integrating exercise interventions to manage and prevent the development of musculoskeletal extraintestinal manifestations.

Previous systematic reviews have shown consistently that exercise improves anxiety, depression and cardiorespiratory fitness in IBD [1, 53, 54]. Although the majority of studies in this review are in agreement, one study found no significant improvements in anxiety or depression [27] and the second found no significant improvements in cardiorespiratory fitness [30], both following a combined aerobic and resistance training programme. However, the variation in findings may be explained by the study design with one [27] employing a cross-over trial design with assumptions made that no carryover effects occurred and the second [30] was an uncontrolled pilot study with a small sample size, thus, the study is underpowered to detect a true effect and results should be interpreted cautiously. Studies should focus on employing rigorous methods, with larger sample sizes to reduce the risk of bias and increase the certainty in observations.

Findings in stress, immune parameters and body composition align with previous reviews [1, 26, 53, 54]. However, novel findings also demonstrated significant improvements in body composition measures of body fat and lean tissue mass [27]. This is of particular importance for people with IBD given the metabolic complications faced and should reassure patients and clinicians that exercise can produce favourable improvements and reverse or prevent loss of muscle mass.

Improvements were identified in fatigue, however met with variation depending on the exercise type and duration. Tew and colleagues [28] feasibility HIIT or MICT intervention saw a mean reduction in total fatigue from baseline to 3 and 6 months. However, between-group comparisons were not conducted due to the study's feasibility and underpowered nature to assess efficacy. Similarly, an unsupervised 4-month personalised booklet programme [29] and 12-week aerobic and resistance training intervention [30] saw significant improvements in pre- and post-measurements. However, Jones and colleagues [32] combined impact and resistance training programme saw no significant improvements at 3 months but did at 6 months. The exercise type could explain the variation between studies and the hypothesised theory of the relationship between fatigue and aerobic capacity, with fatigued IBD individuals displaying an impaired aerobic capacity and muscular strength when compared to non-fatigued IBD individuals [67]. Such a finding suggests participants may benefit from an aerobic exercise programme. In addition, another potential explanation could be the enhanced presence of inflammatory cytokines such as TNF-α, IFN-γ, IL-6, IL-10, IL-12 present in fatigued IBD participants [67]. A recent comparison between an aerobic vs resistance exercise intervention demonstrated significant reductions in TNF-α, IL-6 and IL-10 in the aerobic intervention and no significant differences in the resistance intervention [68]. Suggesting that aerobic exercise may be more appropriate in modulating the immune system, thus symptoms of fatigue, than resistance exercise which may need a longer duration in order to elicit these improvements, which is consistent with findings from this review. Future research is warranted to establish what duration for the type of exercise performed is required to induce improvements in fatigue and evaluate if these are sustained after the intervention period. Overall, there are important methodological limitations future research need to overcome to better understand the exact mechanisms as to how exercise helps fatigue before considering clinical implications. These include high-quality trials that are sufficiently powered, specifically targeted at fatigue and evaluate the different types, durations, frequencies and intensity of exercise on immune parameters.

A key theme for further research should be the involvement of participants with a moderate to severe disease, incorporating support strategies and follow ups examining whether improvements are sustained, cost-effectiveness and comparison of supervised vs unsupervised

interventions. In addition, adverse event reporting needs to be strictly adhered to, as this was lacking in over half of trials.

The strengths of this review benefits from robust methods in keeping with PRISMA and Cochrane guidelines [34, 35] including an overall assessment of physiological and psychological outcomes to give a comprehensive review of the potential benefits of exercise for adults with IBD. The results of this review must be interpreted within the context of their limitations. Although an extensive search of the literature was undertaken using clinical trial registries and major databases, some articles may have been missed due to limiting the search strategy to English, resulting in language and publication bias. The fundamental limitations of the findings in this systematic review with meta-analysis are the heterogeneity of the types of interventions and the outcome measures used to determine the effects of exercise. Moreover, there were no study outcomes with a low risk of bias.

Data in one study [48] was unable to be retrieved, therefore data had to be imputed from studies within the same meta-analysis. As discussed previously, Cochrane guidance could not be followed for one cross-over RCT [27], and therefore was not included in the analysis, which could have impacted results. It is also important to recognise the methodological quality of some of the included articles. Furthermore, the conclusions of this review may have been impacted by the inability to include all studies in a meta-analytic summary, due to a lack of sufficient number of studies for some outcomes. Efforts were made to rectify this, as recommended by the Cochrane Collaboration [35] outcomes that could not be included in the meta-analysis were transparently reported throughout.

This systematic review and meta-analysis found that there was a low certainty evidence that structured exercise interventions of at least 4 weeks' duration reduces disease activity symptoms. There was also very low certainty that there was no clear change between exercise training and usual care in disease specific QOL, however a narrative review suggests exercise may be beneficial. We provide a comprehensive up to date overview of current findings and provide corroborating evidence that support the benefits of exercise in adults with inactive to mildly active IBD. However, results should be interpreted with caution given the heterogeneity of included studies. To date, there are no specific guidelines for clinicians as to what duration, frequency or intensity of exercise is the most appropriate to elicit favourable changes thus it is not surprising that exercise is infrequently discussed by gastroenterologists, GP's and other healthcare professionals. Therefore, there is a need for further large-scale randomised trials to identify the optimal exercise prescription (mode, intensity, duration and frequency) in developing exercise guidelines for adults with IBD. In doing so, patients can be educated about alternative methods of managing and preventing the secondary complications associated with their condition while giving the individual an active role in their treatment. While supporting and educating the CD multidisciplinary team on disease-specific exercise guidelines will enable them to discuss and recommend exercise to optimise health outcomes.

## Supporting information

**S1 Fig. Risk of bias summary plots for meta-analysis.**
(TIF)

**S1 Table. Deviations from registered protocol.**
(DOCX)

**S2 Table. Ongoing trials.**
(DOCX)

**S3 Table. Risk of bias for all outcomes.**
(DOCX)

**S4 Table. GRADE approach.**
(DOCX)

**S1 File. Types of outcome measures.**
(DOCX)

**S2 File. Search strategy.**
(DOCX)

**S1 Checklist. PRISMA 2020 checklist.**
(DOCX)

**S2 Checklist. PRISMA 2020 for abstracts checklist.**
(DOCX)

## Author Contributions

**Conceptualization:** Katherine Jones, Katherine Baker, Garry A. Tew.

**Data curation:** Katherine Jones, Rachel Kimble.

**Formal analysis:** Katherine Jones, Rachel Kimble.

**Funding acquisition:** Katherine Baker, Garry A. Tew.

**Investigation:** Katherine Jones.

**Methodology:** Katherine Jones, Rachel Kimble, Katherine Baker, Garry A. Tew.

**Project administration:** Katherine Jones.

**Resources:** Katherine Jones, Rachel Kimble.

**Software:** Katherine Jones, Rachel Kimble.

**Supervision:** Katherine Baker, Garry A. Tew.

**Validation:** Katherine Jones, Rachel Kimble, Katherine Baker, Garry A. Tew.

**Visualization:** Katherine Jones, Rachel Kimble, Katherine Baker, Garry A. Tew.

**Writing – original draft:** Katherine Jones.

**Writing – review & editing:** Katherine Jones, Rachel Kimble, Katherine Baker, Garry A. Tew.

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
