## [Decision Letter · Decision Letter 0]

30 Aug 2022

PONE-D-22-06927Effects of structured exercise programmes on physiological and psychological outcomes in adults with inflammatory bowel disease (IBD): a systematic review and meta-analysis

PLOS ONE

Dear Dr. Katherine Jone,

Thank you for submitting your manuscript to PLOS ONE. After careful consideration, we feel that it has merit but does not fully meet PLOS ONE’s publication criteria as it currently stands. Therefore, we invite you to submit a revised version of the manuscript that addresses the points raised during the review process.

ACADEMIC EDITOR:

Dear Dr. Katherine Jone,

Your paper "Effects of structured exercise programmes on physiological and psychological outcomes in adults with inflammatory bowel disease (IBD): a systematic review and meta-analysis" has now been reviewed by outside reviewers and me.

As the authors stated in the Introduction, steroid usage has a significant effect on bone mineral density and muscle dysfunction. Can these factors be analyzed in detail?

Also, as mentioned in the Limitation section, there is so much heterogeneity in the type of intervention (exercise method) that it makes me wonder about the meaning of integration.

The biggest problem is that although PROSPERO states that it will conduct a systematic review of muscle and bone problems, it conducts reviews of other factors such as disease activity, quality of life, and fatigue, and does not conduct the essential analysis of muscle and bone problems. The problem is that the analysis and discussion on muscle and bone has became shallow. The analysis should focus on the primary outcome in accordance with the protocol, and a major revision of the protocol is needed.

As there are some errors, the authors should carefully check the text.

Yours sincerely

Shintaro

Shintaro Sagami

Academic Editor

We look forward to receiving your revised manuscript.

Kind regards,

Shintaro Sagami

Academic Editor

PLOS ONE

https://journals.plos.org/plosone/s/file?id=ba62/PLOSOne_formatting_sample_title_authors_affiliations.pdf".

“This work was funded through PhD studentship at Northumbria University.”

“This work was funded through PhD studentship at Northumbria University. The funders had no role in study design, data collection and analysis, decision to publish, or preparation of the manuscript.”

“KJ, GT and KB were investigators on one if the included trials (Jones et al., 2020), however we do not believe that this has biased our assessment of this or any other study.”

Reviewers' comments:

Reviewer's Responses to Questions

**Comments to the Author**

1. Is the manuscript technically sound, and do the data support the conclusions?

Reviewer #1: Yes

Reviewer #2: Yes

2. Has the statistical analysis been performed appropriately and rigorously? 

Reviewer #1: Yes

Reviewer #2: Yes

3. Have the authors made all data underlying the findings in their manuscript fully available?

Reviewer #1: Yes

Reviewer #2: Yes

4. Is the manuscript presented in an intelligible fashion and written in standard English?

Reviewer #1: Yes

Reviewer #2: Yes

5. Review Comments to the Author

Reviewer #1: The manuscript is well written and presented. However, I have several suggestions that I would like the authors to consider.

Introduction:

1. Line 67-88

I find that the sentences explaining general information about IBD is too long (line 67-88) and not sufficiently informative for this review, thus authors need to be shortened. Instead, it would be helpful to be more focused on the review topic. E.g. how the intervention might work; mechanisms that structured exercise programmers effects on physiological and psychological outcomes in IBD.

Method:

2. Line 115

Did the authors include cluster-RCTs and cross-over RCTs, as well as parallel RCTs in this review? If so, how authors handled (or planned) the unit-of-analysis issues for those different designed RCTs.

3. Line 125

Please commenting on how authors handled study that reported outcome on multiple measures (e.g., both IBDQ and SF-36). Moreover, commenting on which outcome measures were prioritized would have been helpful.

4. Line 132

The search seems comprehensive, but is now 1.5 years old (March 2021). It needs to be updated the search to the recent date and revised the manuscript and flow diagram.

5. Line 213

The ROB 2.0 tool produces judgments for individually-randomized, parallel RCTs, but it does not for cluster-RCT or cross-over design. Authors need to use the ROB 2.0 variants for them according to the Cochrane method.

6. Line 217

Please comment how to judge the ‘overall risk of bias’ in this review. Is this corresponded to the worst domain bias judgment?

Results:

7. Line 519

The results of GRADE assessment were about quality of evidence, thus should not be included in the Risk of Bias section.

Other comments:

8. Line 517, Line 520

Supplementary materials in the manuscript are incorrectly numbered. (Supplementary material 7, 8)

Reviewer #2: It is my pleasure to review the manuscript entitled "Effects of structured exercise programmes on physiological and psychological outcomes in adults with inflammatory bowel disease (IBD): a systematic review and meta-analysis". The manuscript is generally well written. However, I have several concerns that need to be addressed before considering publication.

1) Please correct the spelling in the Supplementary Material Content.

"anaerobc", "Lympthocytes" may be misspelled.

6. PLOS authors have the option to publish the peer review history of their article (what does this mean?). If published, this will include your full peer review and any attached files.

Reviewer #1: No

Reviewer #2: **Yes: **Sakuma Takahashi

---

## [Author Response · Author response to Decision Letter 0]

24 Oct 2022

Dear Shintaro Sagami

Manuscript No: PONE-D-22-06927

Effects of structured exercise programmes on physiological and psychological outcomes in adults with inflammatory bowel disease (IBD): a systematic review and meta-analysis

We would like to thank the editor and reviewers for their time reviewing our manuscript and their constructive feedback. We have considered each comment in turn and provide a detailed response to each on the following pages. We have also attached an updated version of our manuscript changes highlighted in Track Changes.

As per requested please see below the updated sections on funding and competing interests to be updated in the online submission:

Funding

This work was funded through PhD studentship at Northumbria University. The funders had no role in study design, data collection and analysis, decision to publish, or preparation of the manuscript

Competing Interests 

KJ, GT and KB were investigators on one if the included trials (Jones et al., 2020) and GT in another study (Tew et al., 2019), however we do not believe that this has biased our assessment of this or any other study. This does not alter our adherence to PLOS ONE policies on sharing data and materials.

Academic Editor Comments 

1. As the authors stated in the Introduction, steroid usage has a significant effect on bone mineral density and muscle dysfunction. Can these factors be analyzed in detail? 

The introduction has been updated to reflect the how steroids effect the risk of bone and muscle health. In addition to how exercise can be used to counteract these responses. Whilst we agree it would be good to analyse this in more detail, only one study (Robinson et al., 1998) reported cumulative steroid usage and one reported usage (yes or no) in the last year (Cronin et al., 2019) therefore this was not possible to do. 

2. Also, as mentioned in the Limitation section, there is so much heterogeneity in the type of intervention (exercise method) that it makes me wonder about the meaning of integration.

Whilst we appreciate there is heterogeneity in the type of exercise delivered, these are similar in the fact they are all delivered a structured exercise intervention of more than a 4-week duration, and all employed a usual care group for comparison. Therefore, we feel these are similar enough to be combined meaningfully. Furthermore, the results were combined with a random-effects meta-analysis that estimates different, yet related, intervention effects. Similar to a previous meta-analysis from our research group in PloS One (https://doi.org/10.1371/journal.pone.0262534) the limited research in this domain precludes us from analysing data based on a specific exercise type. While this remains a limitation we do recommend "further well-designed large-scale RCTs are needed to identify the optimal type of exercise prescription". The systematic review provides an up-to-date summary of the available primary research on this topic which we feel will help to inform practice and future research. 

3. The biggest problem is that although PROSPERO states that it will conduct a systematic review of muscle and bone problems, it conducts reviews of other factors such as disease activity, quality of life, and fatigue, and does not conduct the essential analysis of muscle and bone problems. The problem is that the analysis and discussion on muscle and bone has became shallow. The analysis should focus on the primary outcome in accordance with the protocol, and a major revision of the protocol is needed.

Thank you for your comment. We recognise that the registered PROSPERO protocol has some differences to that of the current review and have been fully transparent in reporting these differences on page 8 of the manuscript and in further detail in supplementary material 1. Due to limited available research in this field surrounding bone and muscle health, other outcomes were added as was the inclusion of non-randomised controlled trials. Despite the recent up to date search as part of reviewer 1 comments, this remains the same. We view the broadening of focus as a strength as it provides a more comprehensive overview of outcomes and study designs. 

https://journals.plos.org/plosone/s/file?id=ba62/PLOSOne_formatting_sample_title_authors_affiliations.pdf".

Thank you for providing these links. Formatting has been updated in accordance with the PLOS one style templates and guidance throughout

5. Thank you for stating the following in the Funding Section of your manuscript: “This work was funded through PhD studentship at Northumbria University.” We note that you have provided funding information that is not currently declared in your Funding Statement. However, funding information should not appear in the Acknowledgments section or other areas of your manuscript. We will only publish funding information present in the Funding Statement section of the online submission form. Please remove any funding-related text from the manuscript and let us know how you would like to update your Funding Statement. Currently, your Funding Statement reads as follows: “This work was funded through PhD studentship at Northumbria University. The funders had no role in study design, data collection and analysis, decision to publish, or preparation of the manuscript.” Please include your amended statements within your cover letter; we will change the online submission form on your behalf.

The funding statement within the manuscript has been removed and the cover letter updated with the amended statements. 

6. Thank you for stating the following in the Competing Interests section: “KJ, GT and KB were investigators on one if the included trials (Jones et al., 2020), however we do not believe that this has biased our assessment of this or any other study.” Please confirm that this does not alter your adherence to all PLOS ONE policies on sharing data and materials, by including the following statement: ""This does not alter our adherence to PLOS ONE policies on sharing data and materials.” (as detailed online in our guide for authors http://journals.plos.org/plosone/s/competing-interests). If there are restrictions on sharing of data and/or materials, please state these. Please note that we cannot proceed with consideration of your article until this information has been declared.

Many thanks for bringing this to our attention. This statement has now been declared and included in the manuscript on page 41

The competing interests, along with the new declaration has been added to the cover letter to be changed in the online submission 

Reviewer 1 Comments

1. Introduction: Line 67-88

I find that the sentences explaining general information about IBD is too long (line 67-88) and not sufficiently informative for this review, thus authors need to be shortened. Instead, it would be helpful to be more focused on the review topic. E.g. how the intervention might work; mechanisms that structured exercise programmers effects on physiological and psychological outcomes in IBD.

The authors agree and have updated the introduction, detailing the IBD-specific mechanisms that increases the risk of bone and muscle dysfunction in addition to addressing how exercise can counteract these responses 

2. Method: Line 115

Did the authors include cluster-RCTs and cross-over RCTs, as well as parallel RCTs in this review? If so, how authors handled (or planned) the unit-of-analysis issues for those different designed RCTs.

The review did include one cross over RCT (Cronin et al., 2019). We agree this could have been clearer, a line has been added to the manuscript on page 12 to address this. 

3. Method: Line 125

Please commenting on how authors handled study that reported outcome on multiple measures (e.g., both IBDQ and SF-36). Moreover, commenting on which outcome measures were prioritized would have been helpful.

Where studies reported on multiple measures for one outcome, both measures were included. No outcome measures were prioritised. We agree this could have been clearer, and added this line has been added to the manuscript on page 9 to address this

4. Method: Line 132

The search seems comprehensive, but is now 1.5 years old (March 2021). It needs to be updated the search to the recent date and revised the manuscript and flow diagram.

The authors agree the searches have now been re-run. One additional study has been identified and the manuscript, figures and content have been updated to reflect this. 

 and manuscript and figures have been updated. In addition, registers have also been researched and the ongoing studies updated to reflect any changes in the current studies listed in addition to identifying any new studies of interest. 

5. Method: Line 213

The ROB 2.0 tool produces judgments for individually-randomized, parallel RCTs, but it does not for cluster-RCT or cross-over design. Authors need to use the ROB 2.0 variants for them according to the Cochrane method.

Many thanks for spotting this error, the ROB cross-over tool has now been used for one study in accordance with the Cochrane guidelines and updated throughout the manuscript and in the supplementary material 

6. Method: Line 217

Please comment how to judge the ‘overall risk of bias’ in this review. Is this corresponded to the worst domain bias judgment?

Many thanks for spotting this omission. A line ‘We derived the overall risk of bias judgement from the highest classified domain i.e. if one domains were classified as high risk this was deemed high risk overall’ has now been added to the ROB section on page 13 detailing on overall risk of bias was judged. 

7. Results: Line 519

The results of GRADE assessment were about quality of evidence, thus should not be included in the Risk of Bias section.

This has been updated to reflect a certainty of evidence section in the methods on page 12 results on page 33

8. Supplementary Material: Line 517, Line 520

Supplementary materials in the manuscript are incorrectly numbered. (Supplementary material 7, 8)

Many thanks for identifying this error, this has now been rectified 

Reviewer 2 Comments: 

1. Please correct the spelling in the Supplementary Material Content. "anaerobc", "Lympthocytes" may be misspelled.

Many thanks for spotting these, a full proofread has been undertaken and grammatical errors corrected.

---

## [Decision Letter · Decision Letter 1]

17 Nov 2022

Effects of structured exercise programmes on physiological and psychological outcomes in adults with inflammatory bowel disease (IBD): a systematic review and meta-analysis

PONE-D-22-06927R1

Dear Dr. Jones,

We’re pleased to inform you that your manuscript has been judged scientifically suitable for publication and will be formally accepted for publication once it meets all outstanding technical requirements.

Kind regards,

Shintaro Sagami

Academic Editor

PLOS ONE

Additional Editor Comments (optional):

Reviewers' comments:

Reviewer's Responses to Questions

**Comments to the Author**

1. If the authors have adequately addressed your comments raised in a previous round of review and you feel that this manuscript is now acceptable for publication, you may indicate that here to bypass the “Comments to the Author” section, enter your conflict of interest statement in the “Confidential to Editor” section, and submit your "Accept" recommendation.

Reviewer #1: All comments have been addressed

Reviewer #2: All comments have been addressed

2. Is the manuscript technically sound, and do the data support the conclusions?

Reviewer #1: Yes

Reviewer #2: Yes

3. Has the statistical analysis been performed appropriately and rigorously? 

Reviewer #1: Yes

Reviewer #2: I Don't Know

4. Have the authors made all data underlying the findings in their manuscript fully available?

Reviewer #1: Yes

Reviewer #2: Yes

5. Is the manuscript presented in an intelligible fashion and written in standard English?

Reviewer #1: Yes

Reviewer #2: Yes

6. Review Comments to the Author

Reviewer #1: Many thanks to the authors for their comprehensive responses and revision. In my opinion, the clarity of the implication has been improved by them.

Reviewer #2: It is my pleasure to review the manuscript entitled "Effects of structured exercise programmes on physiological and psychological outcomes in adults with inflammatory bowel disease (IBD): a systematic review and meta-analysis" again.

The manuscript has been revised well.

I think this manuscript is now acceptable for publication in PLOS ONE.

7. PLOS authors have the option to publish the peer review history of their article (what does this mean?). If published, this will include your full peer review and any attached files.

Reviewer #1: No

Reviewer #2: **Yes: **Sakuma Takahashi

---

## [Editor Report · Acceptance letter]

21 Nov 2022

PONE-D-22-06927R1 

Effects of structured exercise programmes on physiological and psychological outcomes in adults with inflammatory bowel disease (IBD): a systematic review and meta-analysis 

Dear Dr. Jones:

I'm pleased to inform you that your manuscript has been deemed suitable for publication in PLOS ONE. Congratulations! Your manuscript is now with our production department. 

Kind regards, 

on behalf of

Dr. Shintaro Sagami 

Academic Editor

PLOS ONE